# Correlates of the HIV-associated neurocognitive disorders among adults living with HIV in Dodoma region, central Tanzania: A cross-sectional study

**Azan A. Nyundo** *

Department of Psychiatry and Mental Health, School of Medicine, The University of Dodoma, Dodoma, Tanzania

* azannaj@gmail.com, azan.nyundo@udom.ac.tz

## Abstract

### Background

HIV-associated neurocognitive disorders (HAND) continue to manifest despite advancements and improved antiretroviral therapy coverage. Neurocognitive impairment is a significant predictor of poor prognosis related to poor antiretroviral therapy adherence and retention in HIV care.

### Methods

This cross-sectional study examined 397 participants attending cared for and treatment at Dodoma Regional Referral Hospital (DRRH) and selected by systematic sampling. The combination of Montreal Cognitive Assessment (MoCA), International HIV Dementia Scale (IHDS), and The Lawton Instrumental Activity of Daily Living (IADL) were used to assess HIV-associated neurocognitive disorders. Factors associated with HAND were determined using univariate and multivariable logistic regression.

### Results

Of 397 participants, 234(59.1%) met the criteria for HAND with 231(58.2%) comprising asymptomatic neurocognitive disorder (ANI) or mild neurocognitive disorders (MND), and 3 (0.76%) HIV- associated dementia (HAD). Participants with HAND had significantly poorer performance in each cognitive domain on both MoCA and IHDS. Under multivariable regression, age of 55 years or above with Adjusted Odds Ratio (AOR): 3.5 (95%CI: 1.1, 11.6), p = 0.041 and female gender (AOR): 2.7 (95%CI: 1, 6, 4.5), p<0.001 were significantly associated with HAND. Adherence to antiretroviral therapy AOR: 0.4(95%CI: 0.2, 1.0), p = 0.044, and attaining primary education AOR: 0.3(95%CI: 0.1, 0.8), p = 0.01 or secondary education AOR: 0.1(95%CI: 0.03, 0.2), p<0.001 compared to having no formal education showed good cognitive performance.

**Data Availability Statement:** All relevant data are within the paper and its Supporting Information files.

**Funding:** Yes- The author A.A.N received funding for data collection process for the study from Fogarty International Center of the National Institutes of Health "https://www.fic.nih.gov/" under award number D43TW009775. The funders played no role in the study design, data collection and analysis, decision to publish, preparation of the manuscript or supporting for publication processing charges.

**Competing interests:** No-authors have no competing interests

## Conclusion

HIV-associated neurocognitive disorders are common in HIV, especially ANI and MND, are common in HIV infected Tanzanians. Both socio-demographic and clinical variables influence neurocognitive functioning in this population. Screening for mild neurocognitive disorders may be indicated if effective treatment becomes available.

## Introduction

HIV-associated neurocognitive disorders (HAND) is a neurological complication attributable to HIV in the central nervous system (CNS), manifesting with deficits in memory, concentration, attention, and motor skills [1, 2]. For clinical and research purposes, FRASCATI criteria are widely used to categorize HAND into Asymptomatic Neurocognitive Impairment (ANI), Mild Neurocognitive Disorders (MND), and a more severe form called HIV-associated disorders (HAD) [3, 4].

Over three decades, the lifespan of people living with HIV (PLWH) has significantly improved with highly active antiretroviral therapy (HAART).

Furthermore, incidence of severe HAD has been reduced by ART; however, nearly half the population continues to present with some form of cognitive impairment within the spectrum of HAND [5].

The brain is the second most affected organ in HIV infection after the lungs; the cascade of events related to the pathogenesis of cognitive impairment commences during the early phases of HIV infection. The virus enters the brain parenchyma [6, 7] through the infected macrophages and lymphocytes or by the passage of cell-free virus and virus release from the infected endothelial cells [8]. The absence of overt neuropathologies that were common before the cART era suggests clinical presentation of HAND is attributed to functional alteration of neuronal connectivity as opposed to gross neuronal loss or encephalitis [9, 10].

HAND remains prevalent in the cART era, albeit with less advanced stages of HIV infection or severe neurocognitive impairment. Currently, the less severe ANI and MND are observed even among patients with high CD4+ count and undetectable viral load as opposed to pre-ART era, where low CD4+ count, low weight, and anemia were the significant risks for HAND [11]. Nadir CD4+ count, advanced age, hypertension, and dyslipidemia, hepatitis C co-infection (HCV), type of ART used, and psychosocial factors including depression, anxiety, and stigma also remain risk factors for HAND [12–15].

The interaction of sociodemographic and clinical factors adds to the complexity of understanding the course of HAND and offers challenges in diagnostic and designing intervention. This study therefore, examines with standard quantitative methods the prevalence of and risk factors for HAND among adult patients on ART attending an HIV specialty care clinic in the Dodoma region of Tanzania.

## Materials and methods

### Study design

This is a hospital clinic-based prospective observational cross-sectional, single site study.

### Study settings

Dodoma is the capital of Tanzania, located in the central part of the country, with about 410,956 people as per the 2012 census [16]. The study was conducted in the Dodoma region's

main referral hospital, "Dodoma Regional and Referral Hospital" (DRRH). Care and treatment clinic (CTC) services are also conducted in the hospital five days a week, covering up to 100 patients a day coming mostly from Dodoma, and few come outside of Dodoma. Other services provided at the CTC include ART medications, HIV-related counseling services, routine investigations such as CD4+ count, viral load, general medical care, and medical referrals in case of need.

## Study population and sampling

During the study period from March to June 2020, the CTC had enrolled10, 288 patients on ART, of whom 3,708 were available for participation. Using a Kish Leslie formula n = $Z^2$p (1-p) /d2 for a single proportion, a sample size (n) of 384 was calculated, P estimated at 0.5. Z = 1.96, d = 0.05. A final sample size of 397 was attained through a systematic sampling procedure whereby attendance was used to identify and direct every third participant for an interview after inclusion and exclusion criteria were applied.

## Inclusion/Exclusion criteria

We included all patients at least 18 years of age and being on ART for a minimum of six months. The participants must be able to provide informed consent with adequate vision, hearing, articulation, and without disability of any of the upper limbs for neurocognitive assessment. Those who could not read and write in English and Swahili were excluded from the study. We also excluded those with active CNS infection, known complications of previous CNS infection, neurological disorders, in active phase of a psychotic episode and also those with comorbid cardiometabolic conditions including hypertensive disorders, diabetes mellitus.

## Data collection, variables, and measurements

**Outcome variable.** *HIV-associated neurocognitive disorders*. The neurocognitive assessment was determined using the Swahili-translated Montreal Cognitive Assessment (MoCA) and International HIV dementia scale (IHDS).

A meta-analysis found that a cutoff point of 23/30 on MoCA has better diagnostic accuracy across all domains than the routinely used 26/30 cutoff score, especially in a population with less education [17]. The Swahili version of MoCA has acceptable reliability, a sensitivity of 70% and specificity of 60% for MCI and sensitivity of 72% and specificity for dementia in the rural Dodoma older adult population [18]. Since the population used was homogeneously made of older adults from the rural population without HIV, the results from this study were not used as comparative or control group for our relatively younger HIV population (mean age + 41 years). Nonetheless, the population could still demonstrate the effectiveness of the tools in an African population with low education. The instrument also has a receiver operating characteristic curve with an area under the curve of 0.71; MoCA has been widely used worldwide among people with or without symptomatic HAND [19] and is considered a practical screening tool for cognitive assessment in HIV-population [20].

HIV-associated neurocognitive impairment (HAND) was assessed using the combination of the score of Montreal Cognitive Assessment (MoCA), the International HIV-associated dementia scale (IHDS), and The Lawton Instrumental Activity of Daily Living (IADL). The combination of scores from each instrument provides the following categories.

**No neurocognitive impairment.** Based on MoCA score $\geq$ 23 or IHDS score $\geq$ 10 and IADL score of $\geq$ 8.

**Asymptomatic neurocognitive impairment (ANI)** or **Mild neurocognitive disorder (MND) is** considered positive if the patient scores < 23 in MoCA and a score of <10 on IHDS and IADL score of ≥ 8.

**HIV- Associated Dementia (HAD).** The patient had to score < 23 on MoCA, a score of < 10 on IHDS <10, and an IADL score of < 8.

MoCA assesses six key neurocognitive domains;1) visuospatial-executive with a total of five points comprising of clock drawing, trail making B task, and three-dimension cube copy; 2) naming of unfamiliar animals for a maximum of three points for accurate naming of each animal, language with the maximum of three points attained from sentence repetition for two points and a phonemic fluency task for one point: 3) short-term memory assessing delayed recall of words for a maximum of 5 points; 4) abstraction (verbal abstraction) for a maximum of 2 points; 5) attention and calculation assessing digits forward and backward, target detection using tapping and serial 7s subtraction for a maximum of 6 points; 6) and orientation of time and space for a maximum of 6 points [21].

As for IHDS, three main cognitive domains are assessed; memory recall, motor speed, and psychomotor speeds, each with a maximum score of four points, making a total score of twelve. IHDS has demonstrated its utility in screening for cognitive impairment screening in HIV, with good test-retest reliability and capacity to discriminate between the presence or absence of cognitive impairment 90% of the time [22]. This screening tool has gained a reputation for its good pooled sensitivity of 0.90 [95% confidence interval (CI), 0.88–0.91] and overall specificity of 0.96 (95% CI, 0.95–0.97) under summary receiver operation [23]. Compared to MoCA, which is more sensitive in screening milder forms of HAND, IHDS is more sensitive in screening for severe form of HAND [24].

The Lawton Instrumental activity of daily living (IADL) assessed the functional status of the patient's capacity for self-care, which is usually impaired among patients diagnosed with HIV-associated dementia.

The scale assesses one's ability for daily tasks such as laundry, handling finances, and using a telephone. Measuring eight domains of functioning provides early warning signs of functional decline and can be administered in 10 to 15 minutes. The good performance of IADLs is used as an indicator of corresponding cognitive health status with reliable integrity [25].

**Independent variables.** *Major depressive disorders and substance use and related disorders.* The standardized instrument MINI International Neuropsychiatry Interview Schedule (MINI) was used to assess these variables. MINI has acceptably high validation and reliability scores. It can also be administered much shorter (mean 18.7 + 11.6 min, median 15 min) after a brief training session, clinicians can use it, and lay interviewers require more extensive training [26]. We only used the sub-scale of MDD and substance use disorders in the M.I.N.I because of their direct influence on neurocognitive performance in the HIV population [27–30].

Socio-demographic and clinical profiles were also included as explanatory variables; these include age (in years), marital status, gender, occupation (formal employment or no formal employment), years of formal education, living arrangement, most recent CD4+ count (cells/mm3), current viral load (detectable at ≥ 40 copies/ml), Hepatitis C virus (HCV) infection screening and Hepatitis B virus (HBV), type of ART regimen, HIV/AIDS clinical staging as per WHO criteria, duration of ART use (in years), Hemoglobin (HB) concentration in (mmol/L) and body mass index (BMI measured in Kg/m$^2$.

**Data collection and analysis.** The evidence-based researcher-designed questionnaire collected sociodemographic and clinical information of interest, while psychiatric diagnoses were assessed using MINI. Two bilingual groups were consulted to translate MoCA to Swahili and then back-translated to English while maintaining a similar meaning. Interviews were conducted by trained research assistants who took over forty-five working days to complete the

assessment of 397 participants. The research assistants (RAs) were Medical Doctors by profession working as registrars at a referral psychiatric hospital. The RAs were trained for seven days on how to do the interview, administer and rating the study instruments including MINI, MoCA and IHDS. Pre and post training assessment of RAs was done to assess the level of knowledge and skills gained. Under observed setting, the trained assistants had to demonstrate their competency by correctly interviewing and administer study tools on at least ten HIV patients before commencing the actual data collection. Data were analyzed using SAS version 9.4. Descriptive statistics such as frequency and percentage to describe the categorical variables, while mean and standard deviation (SD) or median and interquartile ranges (IQR) were used for continuous variables and presented using tables and figures where appropriate. Frequencies and proportions were summarized in one decimal place while p-values were summarized into three decimal places (a p-value of <0.001 indicated values reaching this point or below).

A Chi-square test was computed to determine the association between socio-demographic, clinical variables, and neurocognitive impairment. At the same time, a t-test was used to determine the mean difference in neurocognitive scores between participants with HAND versus those without HAND across all cognitive domains for both MoCA and IHDS (See Table 3). Unadjusted binary logistic regression was done for preliminary analysis of factors associated with HAND, Thereafter; variables that reached an overall significance level of < 20% (p-value < 0.2) were computed under multivariable model to adjust for confounders. As most participants were negative for HBV/HCV and used only one type of ART regimen {Tenofovir (TDF) +Lamivudine (3TC) +Dolutegravir (DTG)}, these variables were not included in the logistic regression.

**Ethical considerations and concerns.** The study was approved by the local IRB of the Dodoma University Ethical and Research Committee with reference UDOM/ DRP/134/VOL V/91 once the proposal was fully established and merited approval. Trained research assistants who are medical doctors at the level of registrar provided participants with accurate, detailed information regarding the study. If a participant suffered from a neurological, psychiatric, or any other medical disorder needing urgent treatments, they were referred to specialized care based on the locally agreed protocols. Written informed consent forms were provided for those who could read and write; otherwise witnessed verbal informed consent was to be used as an alternative. No minor was included in the study, if a participant could not consent due to medical or any other reason, a custodian or close relative provided the consent.

## Results

### Socio-demographic and clinical characteristics of the study population by HAND

Out of 397 participants, the majority (69.5%) were females, significantly more males were neurocognitively impairment compared to females (p<0.001). The mean age of the study population was 41.9(12.6) years compared to 43.8(11.8) and 38.8(13.4) years for those with and without HAND, respectively (p<0.001). Other factors significantly associated with HAND included occupation, marital status, and education level (p<0.001). As for the clinical profile, only non-adherence to ART had significantly higher proportions of HAND compared to those with good adherence (p = 0.050), refer (Tables 1 and 2).

**Prevalence of HIV-associated neurocognitive disorders by severity.** Out of three hundred and ninety-seven participants, 59.0% met the criteria for HAND on both MoCA and IHDS, of which 58.2% had either asymptomatic neurocognitive impairment (ANI) or mild neurocognitive impairment (MNI). In contrast, 0.8% had a severe form known as HIV-

**Table 1. Socio-demographic characteristics by HAND status of the study population (N = 397).**

| Variable | All N (%) | HAND N (%) | No HAND N (%) | P-value |
|---|---|---|---|---|
| **Age of the respondent** | | | | <0.001 |
| Mean(SD) | 41.9(12.6) | 43.8(11.8) | 38.8(13.3) | |
| ≤24 | 43(10.8) | 11 (25.6) | 32(74.4) | |
| 25–34 | 65(16.4) | 35(53.9) | 30(46.2) | |
| 35–44 | 104(26.2) | 64(61.5) | 40(38.5) | |
| 45–54 | 117(29.5) | 77(65.8) | 40(34.2) | |
| ≥55 | 68(17.1) | 47(69.1) | 21(30.9) | |
| **Gender** | | | | <0.001 |
| Male | 121(30.5) | 51(42.2) | 70(57.9) | |
| Female | 276(69.5) | 183(66.3) | 93(33.7) | |
| **Marital status** | | | | <0.001 |
| Married/Cohabiting | 162(40.8) | 98(60.5) | 64(39.5) | |
| Never married | 80(20.2) | 26(32.5) | 54(67.5) | |
| Divorced/Separated | 89(22.4) | 68(76.4) | 21(23.6) | |
| Widowed | 66(16.6) | 42(63.6) | 24(36.4) | |
| **Years of formal education** | | | | |
| No formal education | 55(13.9) | 47(85.5) | 8(14.6) | <0.001 |
| Primary complete/incomplete | 245(61.7) | 161(65.7) | 84(84.3) | |
| Secondary education ≥ 8-years | 97(24.43) | 26(26.8) | 71(73.2) | |
| **Occupation status** | | | | |
| Self-employed/unemployed | 350(88.2) | 220(62.9) | 130(37.2) | <0.001 |
| Employed | 47(11.8) | 14(29.8) | 33(70.2) | |
| **Living arrangement** | | | | |
| Alone | 88(22.2) | 50(56.8) | 38(43.2) | 0.806 |
| With nursing staff/caregiver | 71(17.9) | 43(60.6) | 28(39.4) | |
| With spouse | 150(37.8) | 92(61.3) | 58(38.7) | |
| With friends/other | 88(22.2) | 49(55.7) | 39(44.3) | |

associated dementia (HAD), see Fig 1. Furthermore, there were 315(79.4%) and 266(67%) participants who met exclusive criteria of neurocognitive impairment for HIDS and MoCA, respectively, refer (Fig 1).

The mean neurocognitive performance based on MoCA was 19.7 for the study population, while the participants with HAND had significantly lower scores of 16.7 compared to those without HAND, 24.0, p<0.001. The mean neurocognitive performance based on IHDS was 7.9 for the study population, while the participants with HAND had significantly lower scores of 6.9 compared to those without HAND, 9.3, p<0.001. Furthermore, participants with HAND have significantly poorer performance in each cognitive domain on both MoCA and IHDS, refer (Table 3).

Under logistic regression, being female AOR: 2.7(95%CI: 4.5, 1.9), p<0.001 was associated with poor cognitive functioning, while every higher increment in age category was significantly associated with neurocognitive impairment under unadjusted analysis although the association remained significant only for those with 55+ years and above AOR:3.5(95% CI:1.0,11.6), p = 0.04 under adjusted analysis. Adherence to ART AOR: 0.4(95%CI: 0.2, 1.0), p = 0.04, having attained primary education AOR: 0.3(95%CI: 0.1, 0.8), p = 0.01 or secondary education AOR: 0.09(95%CI: 0.03, 0.2), p<0.001 compared to having no formal education were all significantly associated with favorable cognitive performance, refer (Table 4).

**Table 2. Baseline clinical profile and major depressive disorders of the study participants, N(397).**

| Variable | All N (%) | HAND N (%) | No HAND N (%) | Chi-square/Student t-test (p-value) |
|---|---|---|---|---|
| **Major depressive disorders** | | | | 0.176 |
| Yes | 22(5.5) | 16(72.7) | 6(27.3) | |
| No | 375(94.5) | 218(58.1) | 157(41.9) | |
| **Substance use/disorder** | | | | 0.866 |
| Yes | 45(11.3) | 26(57.8) | 19(42.2) | |
| No | 352(88.7) | 208(59.1) | 144(40.9) | |
| **WHO clinical staging** | | | | 0.379 |
| I | 84(21.2) | 43(51.2) | 41 (48.8) | |
| Ii | 154(38.8) | 96 (62.4) | 58 (37.7) | |
| Iii | 115(29.0) | 70 (60.9) | 45 (39.1) | |
| IV | 44(11.1) | 25 (56.8) | 19 (43.2) | |
| **ART regime** | | | | 0.653 |
| TDF + 3TC + DTG | 382(96.2) | 226 (59.2) | 156 (40.8) | |
| Others | 15(3.8) | 8 (53.3) | 7 (46.7) | |
| **Most recent CD4 count** | | | | 0.308 |
| <200 | 53(13.4) | 28 (52.8) | 25 (47.2) | |
| 200–499 | 148(37.3) | 94 (63.5) | 54 (36.5) | |
| ≥500 | 196(49.4) | 112 (57.1) | 84 (42.9) | |
| **Current viral load** | | | | 0.781 |
| Undetected | 88(22.2) | 53 (60.2) | 35 (39.8) | |
| Detected | 309(77.8) | 181 (58.6) | 128 (41.4) | |
| **Adherence to ART** | | | | 0.051 |
| Non-adherence | 41(10.3) | 30(73.2) | 11(26.8) | |
| Adherence | 356(89.7) | 204(57.3) | 152 (42.7) | |
| **Duration of ART use: Mean(SD)** | 6.12(4.6) | 6.08(4.4) | 6.18(4.9) | 0.837 |
| **Haemoglobin(Hb) concentration** | 12.40(2.6) | 12.42(2.5) | 12.38(2.7) | 0.897 |

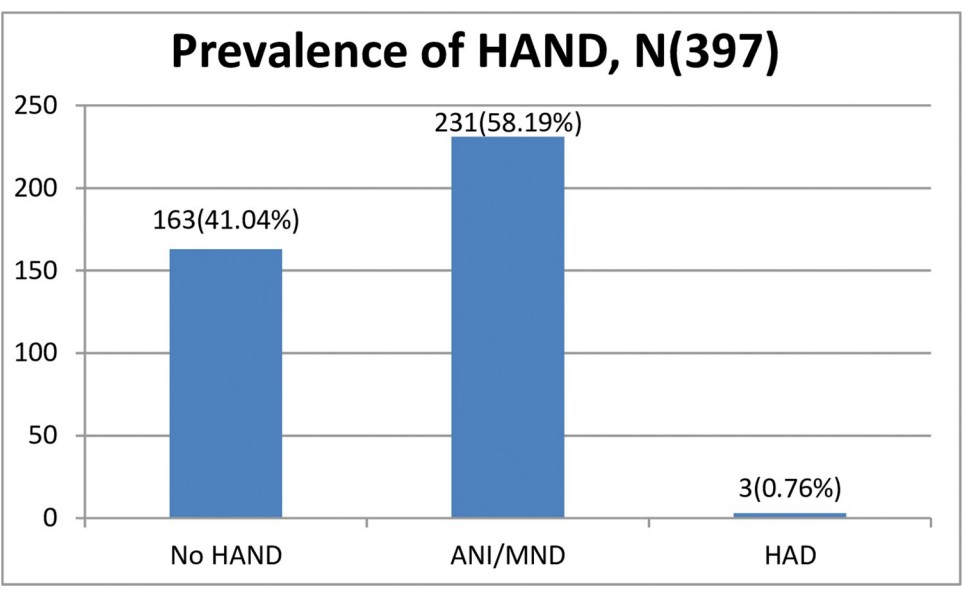

**Fig 1. Prevalence of HAND.**

**Table 3. Pattern of neurocognitive performance by cognitive domain based on MoCA and IHDS, N(397).**

| Domain | All N(397) | HAND (N = 234) | No HAND(N = 163) | P-value |
|---|---|---|---|---|
| **MoCA** | 19.7 (5.2) | 16.6 (4.0) | 24.0 (3.2) | <0.001 |
| Visual executive | 2.7 (1.7) | 1.8 (0.9) | 3.9 (1.3) | <0.001 |
| Naming | 2.3 (0.9) | 2.0 (0.9) | 2.7 (0.6) | <0.001 |
| Attention | 3.3 (1.7) | 2.5 (1.4) | 4.5 (1.4) | <0.001 |
| Language | 0.8 (0.9) | 0.5 (0.7) | 1.2 (0.9) | <0.001 |
| Abstraction | 1.8 (0.5) | 1.7 (0,6) | 1.9 (0.3) | <0.001 |
| Delayed recall | 3.1 (1.5) | 2.5 (1.4) | 3.8 (1.2) | <0.001 |
| Orientation | 5.8 (0.5) | 5.7 (0.6) | 5.6 (0.2) | <0.001 |
| **IHDS** | 7.9 (2.1) | 6.9 (1.7) | 9.3 (1.8) | <0.001 |
| Motor speed | 2.1 (0.8) | 1.9 (0.7) | 2.5 (0.8) | <0.001 |
| Psychomotor speed | 2.7 (1.1) | 2.3 (1.0) | 3.2 (1.0) | <0.001 |
| Memory-recall | 3.1 (1.0) | 2.7 (0.7) | 3.6 (0.7) | <0.001 |

## Discussion

The high prevalence (58.2%) of either asymptomatic or mild neurocognitive impairment that we found in our HIV+ population was associated with advanced age, female sex, less education and ART adherence. With just about 1% met the criteria for HIV-associated dementia; this pattern reflects the effectiveness of HAART in reducing the severe forms of HAND while the prevalence of milder forms remain high, ranging between 21% to 81% [31, 32]. A higher prevalence of neurocognitive disorders in HIV has been reported elsewhere in sub-Saharan Africa. For example, a prevalence of 81.1% was reported from Eldoret, Kenya [24], 68.4% in Dar es Salaam Tanzania [33], and 68.6% in Kampala Uganda [34]. The use of a single screening instrument for neurocognitive function in most of these studies compared to the requirement to meet the criteria on both MoCA and IHDS in our study could explain the difference in prevalence rates. Conversely, a lower prevalence of HAND is reported when a comprehensive neuropsychological battery based on Frascati criteria is used; a meta-analysis shows a global prevalence of HAND to be 44.9%, with the specific a distribution of 26.2%, 8.5%, and 2.1% for ANI, MND and HAD, respectively [35].

Although 41% of the population did not meet the criteria for HAND, the overall mean neurocognitive scores of the whole population were below the cutoff point for IHDS (9.3) but not on MoCA (24.0). Furthermore, the participants with HAND had significantly lower scores than those without HAND across all cognitive domains on both tools, suggesting diffuse brain involvement related to global cognitive impairment. In HIV infection, subcortical brain structures are particularly vulnerable, especially at the early stages [36]; however, the cortical areas are also not spared as the infection progresses and may be linked with motor decline and subsequent cognitive dysfunction [37].

The mean scores of the study population were below the set cut-off point on both IHDS and MoCA; also, more participants (79%) met the criteria for neurocognitive impairment on IHDS compared to 67% on MoCA. This observation may imply the distinct capacity of IHDS to assess specific cognitive domains such as the motor and psychomotor speed that were poorly performed by our participants in general but are not assessed in the MoCA. These specific deficits may suggest an involvement in subcortical structures which are particularly vulnerable in HIV infection and may indicate severe cognitive impairment. Higher proportion of positive screening on IHDS than MoCA may also suggests that the cognitive scores were skewed towards a more severe cognitive impairment within the spectrum of ANI to MND

**Table 4. Logistic regression for factors associated with HAND.**

| | Unadjusted | | Adjusted | |
|---|---|---|---|---|
| Variable | OR: 95%CI | p-value | AOR: 95%CI | p-value |
| **Age** | | <0.001 | | 0.212 |
| ≤24 | Ref | | Ref | |
| 25–34 | 3.4 (1.5,7.9) | 0.004 | 1.7(0.6, 5.0) | 0.357 |
| 35–44 | 4.7(2.1,10.3) | <0.001 | 1.7(0.5, 5.1) | 0.396 |
| 45–54 | 5.6 (2.6,12.3) | <0.001 | 2.3(0.7, 7.1) | 0.167 |
| ≥55 | 6.5(2.8,15.3) | <0.001 | 3.5(1.1,11.6) | 0.041 |
| **Gender** | | | | |
| Male | Ref | | | Ref |
| Female | 2.701(1.7,4.2) | <0.001 | 2.674(1.588, 4.501) | <0.001 |
| **Marital status** | | <0.001 | | 0.065 |
| Married/Cohabiting | Ref | | Ref | |
| Never married | 0.3 (0.2, 0.6) | <0.001 | 0.708 (0.317, 1.584) | 0.401 |
| Divorced/Separated | 2.1 (1.2, 3.8) | 0.011 | 1.694 (0.889, 3.231) | 0.109 |
| Widowed | 1.1 (0.6, 2.1) | 0.659 | 0.614 (0.299, 1.258) | 0.183 |
| **Years of formal education** | | <0.001 | | |
| No formal education | Ref | | Ref | |
| Primary complete/incomplete | 0.326(0.147, 0.722) | 0.006 | 0.333 (0.144, 0.770) | 0.01 |
| Secondary education ≥ 8-years | 0.062(0.026, 0.149) | <0.001 | 0.091(0.035, 0.238) | <0.001 |
| **Occupation** | | <0.001 | | 0.399 |
| Self-employed/unemployed | Ref | | Ref | |
| Employed | 0.251(0.129, 0.486) | | 0.694(0.297, 1.621) | |
| **Current living arrangement** | | 0.806 | | |
| Lives alone | Ref | | | |
| Lives with nursing staff/caregiver | 1.167(0.618, 2.205) | 0.634 | | |
| Lives with spouse | 1.206 (0.706, 2.058) | 0.493 | | |
| Lives with friends/other | 0.955 (0.526, 1.732) | 0.879 | | |
| **Current MDD** | | 0.183 | | 0.608 |
| Yes | 1.919(0.735, 5.015) | | 1.358(0.422,4.372) | |
| No | Ref | | Ref | |
| **Substance use/disorder** | | 0.867 | | |
| Yes | 0.947(0.505, 1.776) | | | |
| No | | | | |
| **Current WHO staging** | | 0.383 | | |
| Stage I | Ref | | | |
| Stage II | 1.578 (0.922, 2.702) | 0.096 | | |
| Stage III | 1.483 (0.840, 2.619) | 0.174 | | |
| Stage IV | 1.255 (0.602, 2.614) | 0.545 | | |
| **Most recent CD4+ count** | | 0.309 | | |
| <200 | Ref | | | |
| 200–499 | 1.554(0.824, 2.932) | 0.173 | | |
| ≥500 | 1.190(0.647, 2.189) | 0.575 | | |
| **Current viral load** | | 0.782 | | |
| Undetected | Ref | | | |
| Detected | 1.071 (0.660, 1.736) | | | |
| **Adherence to ART** | | | | |
| Optimal adherence | 0.492(0.239, 1.013) | 0.054 | 0.401(0.164, 0.977) | 0.044 |

(*Continued*)

**Table 4.** (Continued)

| | Unadjusted | | Adjusted | |
|---|---|---|---|---|
| Sub-optimal adherence | Ref | | | |
| **Duration of ART use** | 1.004(0.961,1.049) | 0.852 | | |

regardless of just about 1% meeting the criteria for most severe form of HAND known as HIV associated Dementia [24].

Regression analyses found that being female and older were significantly associated with neurocognitive impairment. Women had poor cognitive functioning compared to male, especially in speed of information processing (SIP), memory, and motor functions [38]. The sex difference in neurocognitive impairment is attributed to cognitive reserve, mental health, and other comorbidities and biological factors [38].

The cognitive reserve is a factor in determining neurocognitive impairment when the brain is traumatized [39, 40]; compared to men, women living with HIV tend to have lower cognitive reserve that is influenced by a sex difference in psychological risk factors which are synergistically or additive to low cognitive reserve prior to HIV infection and set precedence for neurocognitive impairment after the infection [41, 42]. These factors include low education, poverty, depression, barriers to healthcare and early life trauma which are more common among women than men [43, 44].

Stress and early life trauma as well as mental health disorders tend to affect neurocognitive functioning more in women living with HIV than in men; evidence supports that anxiety, depression, PTSD, and perceived stress are all associated with deficits in memory, learning, and attention [45–48].

The gender differences in neurocognitive performance in HIV could also be linked to endocrinological and immunological factors. Generally, sex steroids, including estradiol, progesterone, and testosterone can influence neurocognitive performance in a healthy population and their optimal levels are linked to good verbal performance in females and visual-spatial performance in males [38]. Specific female-related factors, including menstrual cycle, menopause, and pregnancy, bring complex dynamics that influence neurocognitive performance [49] even more. In HIV, through viral suppression and replications, estradiol is directly linked to transcriptional activities that affect neurocognitive activities [50, 51]. Indeed, both progesterone and estradiol impact immunomodulatory mechanisms through chemokine and cytokines, which also influence neurocognitive functioning [52]. Emerging evidence highlights sex differences in immune function and sex-specific genetic determinants of immune response to HIV infection [53] with variation in pathogenesis-related to chronic immune activation linked to HIV-induced neurotoxicity and eventual neurocognitive decline [54, 55]. Although evidence is not well established, the sex difference in monocyte-associated inflammatory biomarkers has been observed. For example, females have higher levels of sCD163 than males [56, 57]. Also, Neopterine, a marker of cellular immune activation whose CSF levels are generally associated with NCI, their plasma concentration appears to be associated with NCI only in WLWH but not in MLWH [58–61].

In this study, age was also significantly associated with poor neurocognitive deficits. While the overall age group was associated with poor cognitive performance at univariate analysis, only those above the age of 55 years were significantly associated with neurocognitive impairment when adjusted for confounding variables. As HIV/AIDS is now considered a chronic disease, the interaction between HIV and aging affects the brain, and neurocognitive functions become more apparent [62]. Our findings support the previous observation that

advanced age is among the factors associated with reduced neurocognitive impairment and vulnerability to HAND in the HIV population [63–66]. The pathogenesis of neurocognitive decline could partly be due to the additive impact of aging and HIV [67, 68]. Emerging evidence also supports the synergistic effect of the two factors: eg, acceleration memory deficits over one year are found among older but not younger population with HIV [69, 70].

The two factors that appeared protective against neurocognitive impairment are higher educational achievement and adherence to ART. Similar to previous studies, a higher level of education was associated with better cognitive performance. Although there were fewer participants with an advanced level of education, it was still evident that the higher a person had, the better the cognitive performance. Advanced education is thought to improve cognitive reserve and delay neurocognitive decline and functional expression of HIV-related neurodegenerative processes [71–73]. Furthermore, the effect of education is reflected when using cognitive screening tools, including MoCA and MMSE, which are sensitive to educational and socioeconomic factors; for this reason, lower cut-off points or the addition of a point is recommended for those patients with less than twelve years of formal education.

As of recent, the relationship between neurocognitive functioning and adherence to ART is increasingly becoming an area of interest for intervention. Given that the ART adherence rate in sub-Saharan Africa is below the recommended 95%, the reciprocal relation between neurocognitive impairment and ART adherence becomes a significant issue of concern. Our study showed that suboptimal adherence was significantly associated with neurocognitive impairment. Other studies have also shown that neurocognitive impairment is a major predictor of poor adherence to ART [74]. However, the temporal relationship between poor adherence and poor cognitive functioning is unclear.

People living with major depression (MDD) had 35% more odds of neurocognitive impairment that those without MDD. Depression is considered an independent risk for cognitive impairment, indeed, the analysis of the same data to determine the influence of MDD on cognition showed depression negatively influences neurocognitive functioning [75].

The study had several limitations; first, being a cross-sectional study limits its capacity to delineate the temporal relationship between HAND and the explanatory variables. Secondly, using MoCA and IHDS instead of the gold standard comprehensive neuropsychological assessment lowers the diagnostic accuracy and ability to fully categorize HAND into ANI, MND, and HAD. However, both MoCA and IHDS are recommended instruments in settings where the use of ideal gold standard assessment is not feasible. Furthermore, to improve the sensitivity and specificity, the HAND diagnosis was only met if both MoCA and IHDS scores were below the set cutoff points. Also, using IADL improved the ability to identify HAD, albeit the inability to distinguish ANI from MND remained a limitation.

## Supporting information

**S1 Data.**
(SAV)

## Acknowledgments

The author acknowledges the Dodoma regional Referral Hospital for the permission to conduct the study in the hospital premises. Sincere gratitude goes to Dr Sadiki Mandari, Dr Maseto Galikunga, Dr Amina Ally and Dr Joshua John for their impressive work as research assistants. Last but not least, special appreciation to the clients attending the CTC clinic for their corporation and willingness to participate.

## Author Contributions

**Conceptualization:** Azan A. Nyundo.

**Data curation:** Azan A. Nyundo.

**Formal analysis:** Azan A. Nyundo.

**Funding acquisition:** Azan A. Nyundo.

**Investigation:** Azan A. Nyundo.

**Methodology:** Azan A. Nyundo.

**Writing – original draft:** Azan A. Nyundo.

**Writing – review & editing:** Azan A. Nyundo.

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
