## [Decision Letter · Decision Letter 0]

7 Sep 2022

PONE-D-22-19422Correlates of the HIV-associated neurocognitive disorders among adults living with HIV in Dodoma region, central Tanzania: a cross-sectional studyPLOS ONE

Dear Dr. Nyundo,

Thank you for submitting your manuscript to PLOS ONE. After careful consideration, we feel that it has merit but does not fully meet PLOS ONE’s publication criteria as it currently stands. Therefore, we invite you to submit a revised version of the manuscript that addresses the points raised during the review process.

The manuscript has been evaluated by two reviewers, and their comments are available below.

The reviewers have raised a number of concerns that need attention, and they request additional information on methodological aspects of the study and analyses as well as overall improvement in the reporting and presentation. Please note that further consideration will depend on satisfactorily addressing the identified issues. 

Could you please revise the manuscript to carefully address the concerns raised?

We look forward to receiving your revised manuscript.

Kind regards,

Vanessa Carels

Staff Editor

PLOS ONE

Journal Requirements:

"No-authors have no competing interests"

Reviewers' comments:

Reviewer's Responses to Questions

**Comments to the Author**

1. Is the manuscript technically sound, and do the data support the conclusions?

Reviewer #1: Yes

Reviewer #2: No

2. Has the statistical analysis been performed appropriately and rigorously? 

Reviewer #1: I Don't Know

Reviewer #2: I Don't Know

3. Have the authors made all data underlying the findings in their manuscript fully available?

Reviewer #1: No

Reviewer #2: Yes

4. Is the manuscript presented in an intelligible fashion and written in standard English?

Reviewer #1: No

Reviewer #2: Yes

5. Review Comments to the Author

Reviewer #1: The paper addresses HAND, an important long-term consequence of HIV that impacts on its management and has not been widely addressed with adequate methods in Africa. The writing is inadequately precise, concise and clear. I have edited the paper in the PDF "edit" mode and attached the document to this review in hopes it can help the authors with a revision.

While the contents of the methods, results, analysis and discussion are generally good. A number of issues require attention from the authors:

Methods /Analysis

1) The study design was not well described.

2) They use two standardized instruments to assess cognition that have been used in other studies in Africa and use cutoffs that appear appropriate or less educated populations, but do not discuss why they did not use results of a prior study of cognition in the general rural population (ref 18) to assess their participants performance.

3) They did not discuss the details of the educational levels, training and assessment of competence of the interviewers/ testers.

4) They assess and report bivariate analysis of risk factors for HAND, but do not appear to have performed multivariate modeling of there factors which are likely to be correlated.

Such as analysis is standard for this sort of study.

Results/ discussion

5) Several times the authors refer to cognitive "decline" (implying a change from baseline to follow-up) when they mean "impairment" (less than expected normative performance for the population) in this cross sectional study.

6) Data in results is reported to to more decimal places (4) than is justified by either the precision of the estimate or needs of the reader.

7) The paragraph in the discussion beginning with "PLWH with major depresson (MDD)..." is not clear and needs revision.

Reviewer #2: The authors tested neurologic performance and partially describe a representative cohort from their HIV care system. Their techniques use the combination of two screening tests and an ADL assessment to categorize the subjects with regard to HAND. There is no control population without HIV, and the associationof the deficit with HIV is inferential.The status of medical issues, cardiovascular risk and the like which increasingly are associated with poorer performance of HIV populations is incompletely reported. The population is of interest since they have access to more effective therapy than many African populations previously studied. However, the interpretation of the findings is impossible in a cross sectional and uncontrolled use of tests not designed to achieve a secure diagnosis of HAND.

Associations with poor performance include advancing age and females are likely much over-interpreted in the discussion.

Protective factors of education and compliance with treatment are expected.

While the project represents a great effort, the manuscript attempts to over interpret it. Given emerging association with cardiovascular risk factors, and co-morbidities, more emphasis on analysis of these may be also helpful.

Table 4 is hard to follow.

6. PLOS authors have the option to publish the peer review history of their article (what does this mean?). If published, this will include your full peer review and any attached files.

Reviewer #1: **Yes: **J Allen McCutchan

Reviewer #2: No

---

## [Author Response · Author response to Decision Letter 0]

18 Oct 2022

Comments are attached in the response to reviewers document

---

## [Decision Letter · Decision Letter 1]

20 Apr 2023

PONE-D-22-19422R1Correlates of the HIV-associated neurocognitive disorders among adults living with HIV in Dodoma region, central Tanzania: a cross-sectional studyPLOS ONE

Dear Dr. Nyundo,

Thank you for submitting your manuscript to PLOS ONE. After careful consideration, we feel that it has merit but does not fully meet PLOS ONE’s publication criteria as it currently stands. Therefore, we invite you to submit a revised version of the manuscript that addresses the points raised during the review process.

We look forward to receiving your revised manuscript.

Kind regards,

Dured Dardari, Ph.D

Academic Editor

PLOS ONE

Journal Requirements:

Reviewers' comments:

Reviewer's Responses to Questions

**Comments to the Author**

1. If the authors have adequately addressed your comments raised in a previous round of review and you feel that this manuscript is now acceptable for publication, you may indicate that here to bypass the “Comments to the Author” section, enter your conflict of interest statement in the “Confidential to Editor” section, and submit your "Accept" recommendation.

Reviewer #1: All comments have been addressed

2. Is the manuscript technically sound, and do the data support the conclusions?

Reviewer #1: Yes

3. Has the statistical analysis been performed appropriately and rigorously? 

Reviewer #1: Yes

4. Have the authors made all data underlying the findings in their manuscript fully available?

Reviewer #1: Yes

5. Is the manuscript presented in an intelligible fashion and written in standard English?

Reviewer #1: Yes

6. Review Comments to the Author

Reviewer #1: Second review of author responses to both primary reviews of PONE-D-22-19422R1 by Reviewer 1 4-6-23

Response to reviews

Comments from reviewer # 1

The author has accepted the edits from the PDF of the reviewer # 1 made

Amendments

1. The study design was not well described.

- The study design is now corrected as per reviewer’s suggestions, see

highlighted line number 84.

Please add additional descriptors: “observational, hospital clinic-based”

2. They use two standardized instruments to assess cognition that have been

used in other studies in Africa and use cutoffs that appear appropriate or

less educated populations, but do not discuss why they did not use results

of a prior study of cognition in the general rural population (ref 18) to

assess their participants’ performance.

- The justification of not using the results is given in the operational

description of HAND in the methods section; see line number 120-128. OK

3. They did not discuss the details of the educational levels, training and

assessment of competence of the interviewers/testers.

- This is now discussed in detail; see highlighted line number 198-205. OK

4. They assess and report bivariate analysis of risk factors for HAND, but do

not appear to have performed multivariate modeling of these factors

which are likely to be correlated. Such as analysis is standard for this sort of

study.

- Thank you for this comment. I would like to clarify that the multivariate

modeling was computed; the analysis is now clarified in highlighted in line

number 217-221 and also table 4 showing univariate logistic analysis

(unadjusted) and multivariate/multivariable logistic regression (adjusted)

analysis. OK

5. Several times the authors refer to cognitive "decline" (implying a change

from baseline to follow-up) when they mean "impairment" (less than

expected normative performance for the population) in this cross

sectional study.

- The term now used throughout the document is “impairment” unless citing

references that used the specific term. OK

6. Data in results is reported to more decimal places (4) than is justified by

either the precision of the estimate or needs of the reader.

- With exception to the statistical p-values which are in three decimal

places, while the rest of the numbers were reduced to single decimal place.

The descriptions highlighted in line number 209-212 while changes

are made throughout all the tables. OK

7. The paragraph in the discussion beginning with "PLWH with major

depression (MDD)..." is not clear and needs revision

- The paragraph now made clearer; see line number 406-410.

Comments from reviewers # 2

1. The authors tested neurologic performance and partially describe a

representative cohort from their HIV care system. Their techniques use the

combination of two screening tests and an ADL assessment to categorize

the subjects with regard to HAND.

- I believe this is the background comment that the reviewer did not

expect any response OK

2. There is no control population without HIV, and the association of the

deficit with HIV is inferential.

- The author acknowledges that being a cross sectional study, the

associations made are inferential at best and this is also explained as

limitation in the discussion. Having a control population group would

make this a case-control study which is beyond the scope. OK

3. The status of medical issues, cardiovascular risk and the like which

increasingly are associated with poorer performance of HIV populations is

in completely reported.

- The author agrees with the reviewers that cardiovascular and

comorbidities on neurocognitive disorders have an interaction; however,

since patients with cardiometabolic or cardiovascular disorders were

excluded, the discussion of these factors on HAND was beyond the means

and scope of the study. OK

4. The population is of interest since they have access to more effective

therapy than many African populations previously studied. However, the

interpretation of the findings is impossible in a cross sectional and

uncontrolled use of tests not designed to achieve a secure diagnosis of

HAND.

- The reviewer brings an important limitation as far as cross-sectional study is

concerned; however, the author did not make any attempt to interpret

findings beyond or make any causal relationship between dependent

variable and explanatory. Furthermore, while the reviewer cited the

limitation of the tests used, the author used two different tools to improve

the sensitivity in the diagnosis of HAND. In settings where comprehensive

neuropsychological performance cannot be done, MoCA and IHDS are

the recommended instruments. The combined use of these tools has also

been used elsewhere in Africa; see the study” 10.1186/s12883-020-01857-

3” to improve the sensitivity of assessing HAND. O

5. Associations with poor performance include advancing age and females

are likely much over-interpreted in the discussion. Protective factors of

education and compliance with treatment are expected. While the

project represents a great effort, the manuscript attempts to over interpret

it.

- I am not sure what reviewer’s suggestions are, given there are no

questions or suggestions posed. While the reviewers considered the

discussion was over interpreted, it should be noted that the author did not

make any attempt to show any causal relationship but rather an effort to

describe some of the plausible explanation that could be linked to our

findings. Although, this is a research article from a cross-sectional study, a

thorough literature review is necessary to explore the topic in detail and

develop new related hypothesis. While the protective effect of education

and ART adherence is expected, describing how this may happen is

necessary, also the association between advanced age and female

gender also need to be explored in detail. OK, I agree

6. Given emerging association cardiovascular risk factors, and comorbidities,

more emphasis on analysis of these may be also helpful.

- The author agrees with the reviewers that cardiovascular and

comorbidities on neurocognitive disorders; however, since patients with

cardiometabolic or cardiovascular disorders were excluded, the

discussion of these factors on HAND was beyond the means and scope of

the study. OK

7. Table 4 is hard to follow.

- Table is now made easier to follow. OK

7. PLOS authors have the option to publish the peer review history of their article (what does this mean?). If published, this will include your full peer review and any attached files.

Reviewer #1: **Yes: **J. Allen Mccutchan

---

## [Author Response · Author response to Decision Letter 1]

27 Apr 2023

Response to reviews

The author read all the reviews and it appears that all the reviewers agreed with all the responses from the author with exception to one comment from reviewer number 1 which stated as follows:

- Please add additional descriptors: “observational, hospital clinic-based”

The author accepted reviewer’s suggestion and the study design description is amended as per suggestions, see line number 84.

---

## [Editor Report · Decision Letter 2]

2 May 2023

Correlates of the HIV-associated neurocognitive disorders among adults living with HIV in Dodoma region, central Tanzania: a cross-sectional study

PONE-D-22-19422R2

Dear Dr. Nyundo,

We’re pleased to inform you that your manuscript has been judged scientifically suitable for publication and will be formally accepted for publication once it meets all outstanding technical requirements.

Kind regards,

Dured Dardari, Ph.D

Academic Editor

PLOS ONE
---

## [Editor Report · Acceptance letter]

17 May 2023

PONE-D-22-19422R2 

Correlates of the HIV-associated neurocognitive disorders among adults living with HIV in Dodoma region, central Tanzania: a cross-sectional study 

Dear Dr. Nyundo:

I'm pleased to inform you that your manuscript has been deemed suitable for publication in PLOS ONE. Congratulations! Your manuscript is now with our production department. 

Kind regards, 

on behalf of

Dr. Dured Dardari 

Academic Editor

PLOS ONE